# Micro RNA Dysregulation in Keratinocyte Carcinomas: Clinical Evidence, Functional Impact, and Future Directions

**DOI:** 10.3390/ijms25158493

**Published:** 2024-08-03

**Authors:** Jessica Conley, Benjamin Genenger, Bruce Ashford, Marie Ranson

**Affiliations:** 1Molecular Horizons, School of Chemistry and Molecular Bioscience, University of Wollongong, Wollongong, NSW 2500, Australia; jrc739@uowmail.edu.au (J.C.); bg038@uowmail.edu.au (B.G.); 2Illawarra Shoalhaven Local Health District (ISLHD), NSW Health, Wollongong, NSW 2500, Australia; bruceash@uow.edu.au; 3Graduate School of Medicine, University of Wollongong, Wollongong, NSW 2500, Australia

**Keywords:** cutaneous squamous cell carcinoma, basal cell carcinoma, non-melanoma skin cancer, keratinocyte carcinoma, miRNA, micro RNA

## Abstract

The keratinocyte carcinomas, basal cell carcinoma (BCC), and cutaneous squamous cell carcinoma (cSCC), are the most common cancers in humans. Recently, an increasing body of literature has investigated the role of miRNAs in keratinocyte carcinoma pathogenesis, progression and their use as therapeutic agents and targets, or biomarkers. However, there is very little consistency in the literature regarding the identity of and/or role of individual miRNAs in cSCC (and to a lesser extent BCC) biology. miRNA analyses that combine clinical evidence with experimental elucidation of targets and functional impact provide far more compelling evidence than studies purely based on clinical findings or bioinformatic analyses. In this study, we review the clinical evidence associated with miRNA dysregulation in KCs, assessing the quality of validation evidence provided, identify gaps, and provide recommendations for future studies based on relevant studies that investigated miRNA levels in human cSCC and BCC. Furthermore, we demonstrate how miRNAs contribute to the regulation of a diverse network of cellular functions, and that large-scale changes in tumor cell biology can be attributed to miRNA dysregulation. We highlight the need for further studies investigating the role of miRNAs as communicators between different cell types in the tumor microenvironment. Finally, we explore the clinical benefits of miRNAs as biomarkers of keratinocyte carcinoma prognosis and treatment.

## 1. Introduction

Non-melanoma skin cancers are the most common human cancer and the majority (>90%) of NMSC is comprised of the keratinocyte carcinomas (KCs), cutaneous squamous cell carcinoma (cSCC), and basal cell carcinoma (BCC). While BCC is more common (>70% of NMSC) [1], cSCC accounts for 75% of NMSC-related mortality [2]. Due to their common ultraviolet radiation-driven etiology, BCC and SCC share many risk factors including fair skin, age, chronic sun exposure, and hereditary conditions [3,4]. However, the molecular biology of BCC and cSCC differs significantly. Briefly, BCC is driven by aberrant activation of Sonic hedgehog and Smoothened with concurrent loss of PATCHED [5,6]. Conversely, cSCC is typically driven by a mutational inactivation of the tumor suppressors *TP53*, *CDKN2A*, and/or *NOTCH1/2* [3,7]. Subsequent activation of receptor tyrosine kinase signaling (e.g., epidermal growth factor receptor (EGFR)) and/or activation of downstream signaling pathways such as phosphoinositide-3-kinase (PI3K) drives proliferation and tumor growth [3,8,9]. For a more detailed account of the molecular differences between cSCC and BCC, see Refs. [2,10,11].

Micro RNAs (miRNAs) have gained traction as a crucial part of the cellular gene expression regulatory network in development, organogenesis, and pathogenesis since their discovery in 1993 [12]. miRNAs are 19–25 base pairs in length and there are approximately 2600 mature miRNAs [13,14]. The biogenesis of miRNAs has been prominently reviewed by O’Brien et al. [15]. Briefly, the immature pre-miRNA transcript is recognized by the nuclear microprocessor complex (comprising of DGCR8 and DROSHA) and its characteristic cleaved hairpin structure. Upon nuclear export, the terminal loop is removed by the endonuclease Dicer to form the mature miRNA duplex. In the canonical miRNA pathway, miRNAs function as the targeting entity within the RNA-induced silencing complex (RISC). RISC, with assistance of Argonaute proteins, binds to the 3′-untranslated region (UTR) of target mRNAs (in some cases to the 5′UTR region) and causes degradation or translational inhibition based on the level of sequence complementarity to regulate gene expression [12]. In cancer, the miRNA-mediated degradation of tumor suppressor gene transcripts can facilitate cancer growth, invasion, and metastasis. Conversely, oncogene-targeting miRNAs can have tumor suppressive functions [16,17].

Due to their oncogenic and tumor-suppressive functions, miRNAs have been explored as therapeutic targets in cancer with potential for use against KCs. Recent reviews by Menon et al. [18] and Seyhan [19] provide an excellent overview on the strategies for miRNA-targeting therapeutics and its associated challenges. Briefly, strategies targeting oncogenic miRNAs focus on counteracting its regulatory effect on tumor suppressor genes. For this purpose, antagonistic miRNAs, modified nucleic acids, miRNA sponges, masking miRNAs, and small molecule inhibitors targeting miRNA biogenesis are under investigation. Conversely, strategies involving tumor-suppressive miRNAs focus on replenishing miRNA levels by delivering a mimetic or gene therapy. However, there are major challenges associated with the use of miRNA therapeutics (i.e., delivery, stability, and toxicity concerns).

More recent studies also support the use of plasma miRNA levels as therapeutic, diagnostic, and predictive biomarkers [20,21]. Additionally, the use of therapeutic miRNA mimetics and antagonists is being explored to restore tumor-suppressive miRNAs and inhibit oncogenic miRNAs, respectively [22]. The exploration of miRNAs for cancers other than KCs has contributed to a significantly improved understanding of their pathogenesis and progression, and has opened novel therapeutic avenues [18].

Several reviews have summarized the state of miRNA research in KCs in the past [23,24,25,26,27,28]. However, due to the dynamic nature of the field, many have become outdated since their publication over 5 years ago [25,26,27]. More recent reviews have tackled the matter with a broader scope (i.e., epigenetic dysregulation in NMSC or long non-coding RNA dysregulation across all skin cancers) and without critical appraisal of quality of the evidence presented [23,24]. Consequently, there is some overlap between the research presented in this study and these reviews. However, this review reflects a more comprehensive, stringent, and current overview of the field through a clinical lens.

## 2. Methods

The literature was independently searched by two authors (BG and JC) to identify relevant studies for this review using the search terms microRNA, cSCC, and BCC. The search was conducted on PubMed and Google Scholar with no further constraints applied to the search results. Studies researching the role of at least one miRNA in human cSCC or BCC clinical specimens, cell lines, and xenograft models were included. Of the 64 identified studies, 55 covered cSCC, seven covered BCC, and two covered both BCC and cSCC (NMSC). Of note, there were four studies that have been retracted and were consequently excluded from analyses due to credibility concerns. The factors used for assessing quality and reliability are outlined throughout the review (where applicable). No studies were excluded based on quality.

## 3. MicroRNA Dysregulation in cSCC

### 3.1. Consensus of Dysregulated miRNAs in cSCC

A study by Bruegger, Kempf, Spoerri, Arnold, Itin, and Burger [28] published in 2013 investigated the comparative expression of 12 miRNAs in cSCC compared to healthy skin and reported a significant discordance between their findings and those of other related studies. To investigate if the reproducibility issue persists to date, we compiled a list of 140 reportedly differentially regulated miRNAs in cSCC compared to healthy skin (Appendix A) and determined the consensus across studies (Figure 1). Studies with alternate comparisons relating to the progression of cSCC (e.g., metastatic vs. non-metastatic cSCC) are discussed separately in Section 3.3.

We identified several miRNAs that have been reported to be differentially expressed in at least two studies (Figure 1B). The top most consistently reported miRNAs to be upregulated in cSCC were miR-31(-5p) and miR-21-5p. The miR-31 duplex is the immature precursor for miR-31-5p and miR-31-3p. In cell line models of cSCC, miR-31(-5p) mediated increased migration, invasion, and colony formation potentially via reducing RhoBTB1 levels (Table 1) [29,30]. Clinical evidence supports a role of miR-31 in aggressive cSCC and its expression correlates with increased MMP-1 expression, EMT, and micro-vascularization [31]. Future studies should continue to investigate this potentially important role of miR-31(-5p) in cSCC, as it may have as yet undetermined functionality in this disease. For example, miR-31 has well-established tumorigenic roles in other cancers and is linked to multiple additional targets, therapeutic resistance, metastasis, and decreased survival [32,33,34,35]. Furthermore, plasma levels of miR-31 might be an interesting biomarker for aggressive cSCC as it has shown promise in other cancers [34,36,37].

While several studies support the overexpression of miR-21 in cSCC and suggest a link to an invasive phenotype in cSCC, there has been no experimental exploration into a mechanistic understanding for this association [28,38,39,40]. However, the matter warrants closer investigation given the established role of miR-21 in other cancers [41]. A review by Bautista-Sánchez [41] summarizes the utility of miR-21 as a diagnostic and therapeutic biomarker as well as the functional impact of this miRNA in other malignancies. 

The top two miRNAs that are most consistently reported to be downregulated in cSCC are miR-125b and miR-203 (Figure 1B) [38,42,43]. Both miRNAs target genes with known roles in tumor progression. miR-125b has two experimentally confirmed targets in cSCC, *MMP13*, and *STAT3*, and miR-203 targets the proto-oncogene *MYC* (Table 2). Consistent with the canonical functions of its targets, miR-125b reduced invasiveness in cSCC models via MMP13 [38]. Interestingly, the authors report effects on cSCC growth, colony formation, and migration [38]. However, it is more likely that these effects are mediated via an alternate target of miR-125b (i.e., *STAT3*). Additionally, miR-125b reduced viability and cell cycle progression while increasing Bcl2-induced apoptosis via STAT3 [42]. These findings are consistent with changes reported in other cancers. However, evidence collected in other cancers also suggests that the effects of miR-125b on cSCC could be far greater than reported to date [44].

The target of miR-203, *MYC*, has well established roles in tumor growth, keratinocyte differentiation, and metastasis [45,46,47,48]. Accordingly, Lohcharoenkal et al. [43] observed that miR-203 expression correlates with the differentiation status of clinical samples. Transfection of in vitro models of cSCC with miR-203 decreased cell cycle progression, colony formation, migration, and invasion (Table 2). Furthermore, miR-203 inhibited angiogenesis and tumor growth in xenograft models. It is noted, however, that two separate studies reported no differences in expression of this miRNA in cSCC (Figure 1B), calling into question the role of miR-203 in cSCC pathogenesis [28,38].

Thus, while there is some consensus across studies, the many conflicting studies make an unequivocal interpretation of the current literature difficult. We propose three main factors that may explain this inconsistency between studies. Firstly, the experimental design (including the miRNA extraction method) might introduce a bias by preselecting for highly expressed miRNAs, investigating levels of only one selected miRNA or a predetermined panel of miRNAs (microarrays) in the clinical specimens [49,50]. Secondly, the quality of bioinformatics methodology and the stringency of statistical analysis varies, confounded by methodology changes and improvements over time [49,50]. Lastly, miRNA levels are subject to normal biological differences. For example, factors such as photo age, immunosuppression status, or sex can influence the expression levels of miRNAs [51,52,53,54,55]. Utilizing matched samples can aid in reducing these differences by accounting for the patient-specific background but this approach has not always been used in the literature reviewed here.

### 3.2. Experimentally Validated miRNAs in cSCC

The previous section focused on the consensus, or lack thereof, between all studies that report on the differential expression of miRNAs in clinical specimens. However, miRNA analyses that combine clinical evidence with experimental elucidation of targets and functional impact provide far more compelling evidence than studies purely based on clinical findings or bioinformatic analyses. The following provides a comprehensive overview of studies integrating analysis of miRNA levels in cSCC patient samples, functional investigation of miRNA targets, and integration into larger scale cSCC biology. Table 1 and Table 2 include up- and downregulated miRNAs in cSCC, respectively. The functional effect of the miRNA included in Table 1 and Table 2 represents the changes observed if the miRNA is introduced into the model systems included in the table. The validated target for each miRNA is downregulated in response to the expression of respective miRNA.

**Table 1 ijms-25-08493-t001:** Experimental validation of upregulated miRNA in cSCC compared to healthy skin.

miRNA	Tissue Comparison	Cell or Animal Model	Validated Target	Functional Effect of the miRNA	Ref.
miR-10a	cSCC vs. HS	A431	*SDC1*	Proliferation ↑ Migration, invasion ↑	[56]
miR-10b	cSCC (RDEB and non-RDEB) vs. HS	RDEB-SCC1/2/62 SCC13 A431 WT18SCC	*DIAPH2*	Spheroid formation ↑ Migration ↓ CSC phenotype ↑	[57]
miR-22	cSCC vs. HS	A431 COLO-16 Xenograft	*FOSB* *PAD2*	Migration ↑ EMT, stemness ↑ Spheroid formation ↑ Tumor formation, growth, and metastasis ↑ Wnt/ β-catenin signaling ↑	[58]
miR-31	cSCC vs. HS/ AK	UT-SCC-7	nd	Motility ↑ Migration, invasion ↑ Colony formation ↑	[29]
miR-31-3p	cSCC/IEC/AK vs. HS	COLO-16 SCC9 SCC-25	nd	Viability ↑ Colony formation ↑	[59]
miR-135b	cSCC vs. HS	PM1 MET1 MET4	*LZTS1*	Migration, invasion ↑	[60]
miR-186	cSCC vs. HS	A431 SCL-1	*RETREG1*	Proliferation ↑ Apoptosis ↓	[61]
	cSCC vs. HS	A431	*APAF1*	Apoptosis ↓ Autophagy ↓ Migration, invasion ↑ Colony formation ↑ Cell cycle progression ↑ Proliferation ↑	[62]
miR-217	cSCC vs. HS	SCC13	*PTRF*	Proliferation ↑ Cell cycle progression ↑ Invasion ↑	[63]
miR-221	cSCC vs. HS	A431 SCC13	*PTEN*	Viability ↑ Colony formation ↑ Akt signaling ↑	[64]
miR-320a	cSCC vs. HS	A431 SCL-1 Xenograft	*ATG2B*	Autophagy ↓ Apoptosis ↓ Tumor growth ↑ Proliferation ↑	[65]
miR-346	cSCC vs. HS	A431	*SRCIN1*	Proliferation ↑ Migration ↑	[66]
miR-365	cSCC vs. HS	HaCaT A431 Xenograft	nd	Tumorigenicity ↑ Tumor growth ↑ Colony formation ↑ Migration, invasion ↑ Apoptosis ↓	[67]
	cSCC vs. HS	A431 HSC-1 Xenograft	*NFIB*	Tumor formation and growth ↑	[68]
miR-486-3p	cSCC vs. HS	HSC-5 HSC-1 Xenograft	*FLOT2*	Tumor growth ↑ Viability, proliferation ↑ Migration ↑	[69]
miR-664	cSCC vs. HS	HSC-5 HSC-1 Xenograft	*IRF2*	Tumorigenicity ↑ Migration, invasion ↑ Proliferation ↑	[70]
miR-675	cSCC vs. HS	HaCaT SCL-1 A431	*TP53* *H19*	Proliferation ↑ Migration, invasion ↑ Apoptosis ↓ EMT ↑	[71]
miR-766	cSCC vs. HS	A431 SCL-1 Xenograft	*PDCD5*	Apoptosis ↓ Migration, invasion ↑ Proliferation ↑ MMP2/9 expression ↑ Tumor growth ↑	[72]
miR-7150	cSCC/IEC/AK vs. HS	COLO-16 SCC-9	nd	Viability ↑ Colony formation ↑	[59]

AK—actinic keratosis, HS—healthy skin, EMT—epithelial-to-mesenchymal transition, nd—not determined, IEC—intra-epidermal carcinoma, RDEB—recessive dystrophic epidermolysis bullosa, CSC—cancer stem cell, ↑—increase in phenotype, ↓—decrease in phenotype.

**Table 2 ijms-25-08493-t002:** Experimental validation of downregulated miRNA in cSCC compared to healthy skin.

miRNA	Tissue Comparison	Cell or Animal Model	Validated Target	Functional Effect of the miRNA	Ref.
miR-23b	cSCC vs. HS/ AK	UT-SCC7 UT-SCC12a Xenograft	*RRAS2*	Angiogenesis ↓ Colony formation ↓ Spheroid formation ↓ Tumor growth and proliferation ↓	[73]
miR-31-5p	cSCC/IEC/AK vs. HS	COLO-16 SCC-9	nd	Colony formation ↓	[59]
miR-34a-5p	cSCC vs. HS	A431 SCL-1	*SIRT6*	Proliferation ↓ Colony formation ↓ Migration ↓ Apoptosis ↑	[74]
miR-124	cSCC vs. HS	DJM-1	nd	ERK signaling ↓ Proliferation ↓	[75]
miR-125b	cSCC vs. HS/ AK	UT-SCC-7 A431	*MMP13*	Growth ↓ Colony formation ↓ Migration, invasion ↓	[38]
	cSCC vs. HS	A431 SCC13 SCL-1	*STAT3*	Viability ↓ Cell cycle progression ↓ Apoptosis via Bcl2 ↑	[42]
miR-130a	cSCC vs. HS/ AK	UT-SCC-7 A431 Xenograft	*ACVR1*	HRAS/MAPK signaling ↓ Tumor growth ↓ Tumor sphere formation ↓ Migration, invasion ↓ SMAD1 signaling ↓	[76]
miR-138-5p	cSCC vs. HS	A431 Xenograft	*EZH2*	Autophagy ↓ Apoptosis ↑ Viability ↓ STAT/VERFR2 signaling ↓ Tumor growth ↓	[77]
miR-148a	cSCC vs. HS	A431 SCL-1 Xenograft	*MAP3K4* *MAP3K9*	Colony formation ↓ Proliferation ↓ Migration, invasion ↓ EMT ↓ MAPK signaling ↓ Tumor growth ↓	[78]
miR-181a	cSCC vs. HS	SCC13 Xenograft	*KRAS*	Tumor growth ↓ Viability ↓ ERK signaling ↓	[79]
miR-199a	cSCC vs. HS	A431	*CD44*	Proliferation ↓ Invasion ↓ MMP2/9 expression ↓	[80]
miR-203	cSCC vs. HS	UT-SCC7 A431 Xenograft	*MYC*	Cell cycle progression ↓ Colony formation ↓ Migration, invasion ↓ Angiogenesis ↓ Tumor growth and angiogenesis ↓	[43]
miR-203a-3p	cSCC vs. HS	SCL-1	*APC*	APC/ β-catenin signaling ↓ Proliferation ↓ Colony formation ↓	[81]
miR-204	cSCC vs. AK	HaCaT	*PTPN11*	FGF-STAT3 signaling ↑ EGF-MAPK signaling ↓	[82]
miR-211-5p	cSCC vs. HS	IC4 IC18	*TP63*	Differentiation ↑ EMT ↓ Proliferation ↓	[83]
miR-214	cSCC vs. HS	A431 SCC13	*BCL2* *VEGFA*	Viability ↓ Proliferation ↓ Migration, invasion ↓ Apoptosis ↑ Wnt/ β-catenin signaling ↓	[84]
	cSCC vs. HS	DJM-1	nd	ERK signaling ↓ Proliferation ↓	[75]
miR-340	cSCC vs. HS	A431 Sa3	*RHOA*	Proliferation ↓ Migration, invasion ↓	[85]
miR-342-3p	cSCC vs. HS	A431 SCC13	*NEAT1*	Proliferation ↓ Colony formation ↓ PI3K signaling ↓	[86]
	cSCC vs. HS	SCC13	*SCARNA2*	Proliferation ↓ Cell cycle progression ↓ Invasion ↓	[87]
miR-361-5p	cSCC vs. HS	HaCaT A431	*VEGFA*	*VEGFA* levels ↓	[88]
miR-497	cSCC vs. HS/AK	SCLII MET1	*SERPINE1*	Growth ↓ Migration ↓ EMT ↓	[89]
	cSCC vs. HS	A431 HSC-5	*FAM114A2*	Viability ↓ Cell cycle progression ↓	[90]
miR-1193	cSCC vs. HS	SCC13 COLO-16 Xenograft	*MAP3K9*	Viability ↓ Colony formation ↓ Migration, invasion ↓ Lactate production ↓ Glucose consumption ↓ Tumor growth ↓	[91]
miR-1238-3p	cSCC vs. HS	A431 SCL-1 Xenograft	*FOXG1*	Migration, invasion ↓ Proliferation ↓ Cell cycle progression ↓ Viability ↓ Apoptosis ↑ Tumor growth ↓	[92]

AK—actinic keratosis, HS—healthy skin, EMT—epithelial-to-mesenchymal transition, nd—not determined, IEC—intra-epidermal carcinoma, MMP—matrix metalloproteinase, ↑—increase in phenotype, ↓—decrease in phenotype.

Unsurprisingly, upregulated miRNAs support tumor growth and formation by targeting tumor suppressor genes such as *TP53* or *NFIB* (Table 1). A reduction in tumor suppressor genes by this miRNA targeting enables tumor cell survival and proliferation as well as the acquisition of traits required for metastatic dissemination such as stemness properties, angiogenesis, and invasive capacity in cSCC. Several studies report increased proliferation, viability, and/or decreased apoptosis in in vitro models of cSCC (Table 1). These findings translated to increased tumor formation and growth in xenograft models. Additionally, upregulation of EMT markers and MMPs points towards the capacity of miRNAs to enable disease progression by conferring increased invasive capacity (Table 1). Finally, properties such as increased stemness and angiogenesis might aid in metastatic dissemination of cSCC and warrants investigation of miRNAs as putative biomarkers (see Section 5). Conversely, downregulated miRNAs inhibit oncogenes (e.g., *MYC*, *APC*, *KRAS*) and the aforementioned traits in cSCC such as proliferation, migration, and invasion (Table 2).

While the studies presented in Table 1 and Table 2 highlight the importance of miRNAs in cSCC pathogenesis, it is worth noting that the quality of the evidence presented in these studies varies significantly. We have identified three main variables contributing to the divergent quality of the evidence presented. Firstly, it is crucial to ensure proper validation of the target mRNA that conveys the observed functional changes as miRNAs can have several targets. For this purpose, a luciferase reporter construct (incorporating the 3′-UTR of the mRNA in question), the use of an antisense miRNA or a miRNA mimetic, and knockdown of the protein in question to test for functional equivalence seems to be adequate. Of note, most studies functionally investigate only one potential target, despite compelling bioinformatic evidence of multiple likely targets [93]. Secondly, the choice of the model cell line(s) is important to ensure appropriate representation of the cSCC subtype of interest. For example, A431 is a cell line derived from the vulva and its validity as a model for a UV-induced cancer is questionable [9]. Lastly, the exclusive use of in vitro models inherently limits the translatability of the findings to a complex tumor. Beyond the typical limitation of cell line models, determining the functional impact of miRNAs can be especially challenging as they can be incorporated as cargo in extracellular vesicles (EVs) [94]. Secreted miRNAs are not accounted for when studying tumor cells in monoculture. In other cancers, some evidence suggests that EVs can cause drastic changes in tumor-associated nerves, fibroblasts, and immune cells [95,96]. For example, some evidence suggests that cSCC-derived EVs are taken up by fibroblasts leading to the formation of activated cancer-associated fibroblasts [97,98]. The impact of tumor cell-derived exosomal miRNAs in the cSCC tumor microenvironment (TME) has not yet been investigated and should be considered when assessing the functional impact of miRNAs in cSCC. The use of advanced tissue culture models and xenograft models can help in addressing those questions (Table 2) [99,100] and should be considered in the future.

In cSCC, miRNAs regulate a diverse network of cellular functions and large-scale changes can be attributed to their dysregulation (from basic changes of transformed cells to highly malignant traits of deadly tumors) (Section 3.2). However, the interpretation of the functional impact of miRNAs should be done within a greater context of cSCC biology (i.e., genetic and epigenetic drivers, other transcriptional changes) as well as under special consideration of the interplay of the tumor cells with non-cancerous cell types and the TME.

### 3.3. Differential Expression of MicroRNAs during the Clinical Progression of cSCC

While most of the studies reported above compare miRNA expression between tumor and healthy skin, cSCC exists on a disease progression continuum, which suggests that the dysregulation of the genetic and epigenetic landscape may be an ongoing evolution rather than a well-defined ‘switch’. The stepwise progression from healthy skin through to invasive and metastatic cSCC has been long established and reviewed elsewhere [3,101]. Briefly, the prolonged exposure of skin to UV radiation causes the accumulation of mutations over time, which causes keratinocytes to progress to a precancerous lesion known as actinic keratosis (AK). Acquisition of additional oncogenic mutations can drive AKs towards the cancerous form of cSCC. While there have been significant efforts to define the molecular changes that drive this progression [102,103], the role of miRNA dysregulation remains relatively unclear. Here, we address the growing body of literature that explores miRNA dysregulation in the clinical progression of cSCC.

In an effort to elucidate the role of miRNAs in early keratinocyte pathogenesis, Mizrahi et al. [89] compared total miRNA profiles between five types of lesions—normal skin, solar elastosis (SE), early- and late-stage AKs, and well-differentiated cSCC. While their findings largely support the hypothesis for a gradual progression of disease, they also defined several ‘stage-specific’ alterations. For example, dysregulation of miR-19b and miR-126 was detected early in disease progression, respectively down- and upregulated in SE compared to healthy skin. Conversely, dysregulation of miR-424, miR-378, and miR-497 was detected later in progression, respectively up- and downregulated in cSCC compared to all earlier lesions. Subsequent experimental validation via forced overexpression of miR-497 in two cSCC cell lines demonstrated the role of this miRNA in targeting *SERPINE1* to repress cell growth, migration and EMT phenotype (Table 2). This was, however, the only experimentally validated miRNA of the many identified in this study, and future efforts to characterize these miRNAs may further clarify their role in cSCC progression. Hierarchical clustering of dysregulated miRNAs identified in this study separated the lesions into two main groups—one comprised of normal skin, SE and early-stage AKs, and the other comprised of late-stage AKs and cSCC. This suggests that while there are indeed some changes in miRNA expression early in the disease progression, such as the downregulation of miR-126-5p and let-7i in SE compared to normal skin, most changes in the miRNA expression landscape occur late in the progression from pre-malignant to malignant cSCC.

In another study by Hossain et al. [59], the comparator of intraepidermal carcinoma (IEC) was included to distinguish later stage changes between AK and cSCC. miR-31-5p was found to be upregulated in cSCC compared to IEC, which demonstrates diagnostic potential for late-stage disease. Robust pairwise comparisons between normal skin, photo-damaged skin, AK, IEC, and cSCC were also performed. Of note, miR-7150 was implicated in cSCC for the first time, as it was demonstrated to be upregulated in AK compared to photo-damaged skin, as a discriminative marker for early lesion identification.

Several studies also address the differences in miRNA expression between cSCC lesions with varied differentiation status. miR-340 expression was found to be downregulated in poorly-differentiated compared to well-differentiated cSCC, which was linked to upregulation of its experimentally validated target *RHOA* to promote tumor cell proliferation, migration, and invasion in vitro [85]. Another study found that miR-203 was downregulated in poorly-differentiated tumors compared to both well-differentiated tumors and healthy skin [43]. In this study, in vitro inhibition of miR-203 in keratinocytes reduced calcium-induced differentiation. In addition, forced over-expression of miR-203 in cSCC cell lines led to the downregulation of the oncogene *MYC* and subsequently induced cell cycle arrest, and decreased proliferation, colony formation, migration, invasion, and angiogenesis in vitro. A reduction in tumor growth and angiogenesis was also observed in cSCC xenografts over-expressing miR-203 (Table 2). Conversely, Caneuto et al. [104] demonstrated that miR-203 shows altered expression levels within the same tumor, with significantly higher expression of the miRNA in well differentiated areas of cSCC.

Separately, attempts have been made to clarify the role of miRNAs in driving disease progression in arsenic-induced, RDEB, and organ-transplant recipient etiologies [57,60,105], but these mostly remain unclear. Geusau et al. [55] characterized the tumor, perilesional, and normal skin miRNA profiles of organ transplant recipients. They too demonstrate a stepwise progression pattern of miRNA dysregulation observed in other studies surrounding immunocompetent cSCC patients; however, a comparison between these etiologies is yet to be drawn and further validation is required.

Since lymphovascular invasion (LVI) is associated with poor prognosis and progression of cSCC [106], Robinson et al. [107] investigated the differentially expressed miRNAs between LVI-positive and LVI-negative cSCC tumors. While there were no downregulated miRNAs between groups, miR-155-5p, miR-196a-5p, miR-375, and miR-221-5p were all significantly upregulated in LVI-positive tumors. This may have utility in stratifying patients’ risk status and to identify tumors in need of early intervention.

Although LVI is linked to metastatic risk and this study may provide indirect insight into miRNAs involved in metastasis, very few studies have explicitly investigated the function of miRNAs in cSCC metastasis. Given our understanding of metastatic disease is arguably more clinically informative—since metastases pose the greatest burden and cannot be predicted ahead of time—this was surprising. A recent study by Gillespie et al. [108] compared the expression of approximately 800 miRNAs between metastatic tissue and primary lesions (including both primaries that went on to metastasize and those that did not). There were no differentially expressed miRNAs observed between matched metastatic tumors and the primaries from which they were derived. While this might suggest discrimination of primary and metastatic disease by miRNA expression is not feasible, it is important to note that this study used a pre-defined miRNA panel that might exclude other potentially important miRNAs. It is also possible that these primary tumors that went on to metastasize had already accumulated any pro-metastatic miRNA changes that may exist and hence could not be distinguished between these matched lesions. There were, however, multiple miRNAs differentially expressed between metastatic tumors and all primary tumors (both metastatic and not). Specifically, miR-4286, miR-200a-3p, and miR-148-3p were all upregulated, and miR-1915-3p, miR-205-5p, miR-4515, and miR-150-5p were all downregulated in metastases.

The urokinase plasminogen activator system (uPAS) is a master regulator of MMPs and has previously been implicated in cancer metastasis including in cSCC [9,109,110]. Minaei et al. demonstrated upregulation of the two main components (uPA and uPA receptor). This sparked further investigations into miRNAs potentially targeting *PLAUR* (coding for uPAR). Of the potential candidates, miR-340-5p and miR-377-3p were significantly downregulated in metastatic cSCC compared to both metastasizing and non-metastasizing primary cSCC tumors. Given there was a negative correlation between miR-340-5p expression and uPAR staining in cSCC lymph node metastases, this miRNA-mRNA interaction is strongly implicated in the metastatic process of cSCC. This is further supported by the literature, with downregulation of miR-340 also detected in poorly differentiated cSCC. In addition, miR-497—which targets the gene encoding a uPA inhibitor, *SERPINE1*, or PAI-1—is downregulated in primary cSCC compared to normal skin (Table 2).

Given there are some promising results that elucidate the role of miRNA expression in cSCC progression, further work will benefit from the integrated investigation of genomic, epigenetic, transcriptomic and proteomic drivers of disease, given they are all tightly interconnected and interdependent.

## 4. miRNA Dysregulation in BCC

Much like the state of the literature investigating cSCC, there are few studies describing the patterns and roles of miRNA dysregulation in BCC. Several reviews have summarized what little work has been published elsewhere, though they are now outdated or are not comprehensive [23,24]. The clinical evidence for miRNA dysregulation in BCC is reviewed below.

### 4.1. Consensus of Dysregulated miRNAs in BCC

Given that previous reviews reporting miRNAs dysregulated in BCC are not comprehensive, the concordance of results to date was unclear. To this end, we identified 253 miRNAs with reported dysregulation (Appendix A) and evaluated the consensus between studies (Figure 2). As no miRNAs demonstrating neutral expression across both BCC and controls were explicitly reported, only up- and downregulated miRNAs were included. Further, only comparisons between BCC tissue and healthy skin were assessed.

The marked lack of consensus between studies is evident, with just four miRNAs demonstrating similar trends of dysregulation across more than one study (Figure 2B). Upregulated in BCC compared to healthy skin, miR-941 is a well-established oncogenic miRNA that has been implicated as both a diagnostic biomarker in serum exosomes and as a therapeutic target in other cancers [111]. Furthermore, inhibition of miR-941 has been shown to enhance tumor sensitivity to the chemotherapeutic 5-fluorouracil, highlighting the potential clinical significance of this finding [112]. The downregulated miRNAs—miR-29c and miR-383-5p—are both well-established tumor suppressors in many cancer types. However, though miR-452 was downregulated in BCC here, its upregulation and oncogenic effect in other cancer types indicates its functionality is tumor specific which warrants further investigation in the context of BCC [113].

Overall, there are notably more miRNAs dysregulated in BCC than we reported for cSCC above. While it is possible that this is indeed a biological phenomenon, we speculate that it is a result of the methods employed in the literature. A greater proportion of studies investigating BCC used next-generational sequencing of total small RNA populations, rather than the tailored panels and microarrays that were frequent in investigations of cSCC. Despite these non-biased approaches, which overcome many of the methodological shortcomings observed in the literature surrounding cSCC, there is still discordance among three miRNAs (Figure 2A). This highlights the challenges of comparing biologically diverse specimens, and the complexity of miRNA regulation and indeed BCC pathogenesis.

### 4.2. Experimentally Validated miRNAs in BCC

As mentioned above, the functional effect of miRNA dysregulation is far more informative than isolated reports of differential expression between diseased and healthy samples, as it provides an imperative biological understanding of disease behavior. To this end, we interrogated the literature for studies reporting miRNAs with both clinical dysregulation and functional analysis in BCC (Table 3).

Evidently, the functional characterization of miRNAs in BCC is minimal. This was surprising given that the differential expression analyses were generally of larger scale, identifying more dysregulated miRNAs than most reported in cSCC. We suspect this lack of experimental analysis reflects the clinical nature of BCC; given BCC rarely metastasizes and is generally well managed, the search for therapeutic targets or biomarkers including miRNAs is not of particular urgency [4]. As we observed in the literature regarding cSCC, there was a range in the quality of these functional studies. While both the in vitro and in vivo models for BCC were generally robust, the use of the A431 cell line as a model for BCC is even more questionable than its utility for cSCC. Having been derived from a vulval epidermoid carcinoma, it not only lacks the ultraviolet radiation signature that is characteristic of most skin cancers, but it is also not derived from a basal cell carcinoma at all [117,118]. A431 cells are derived from vulvar SCC and are HPV negative which makes other etiologies such as lichen sclerosis or cellular atypia due to advanced age most likely [119,120]. Interestingly, much of the literature focuses on the dysregulation of circulating miRNAs between BCC and healthy patients, and for this reason were excluded from Table 3, which only addressed miRNAs endogenous to BCC tissue. These are addressed in Section 5.

Only one miRNA—miR-203—demonstrates downregulation with experimental validation across both cSCC and BCC. Though the validated targets differ, with *MYC* and *JUN* targeted in cSCC and BCC, respectively, these are both recurrently altered oncogenes across many cancers. Functionally, forced overexpression of miR-203 demonstrated similar effects in cSCC and BCC, reducing proliferation, cell cycle progression, and tumor growth in cell and xenograft models [43,115]. Given that miR-203 dysregulation has been documented across a range of tumor types [121,122,123], it is unlikely that this observation points to a keratinocyte-specific pathogenesis. Indeed, the lack of consensus surrounding this miRNA in cSCC (Figure 1) complicates the interpretation of this finding without further investigation. The otherwise lack of consensus between miRNAs in cSCC and BCC points to distinctive disease progressions and ultimately highlights the importance of tumor-specific investigation to elucidate the role of miRNAs in KCs.

## 5. Clinical Applications of miRNAs in KCs

While the results summarized above enhance our understanding of the role of miRNAs in KC pathogenesis and progression, they have limited clinical utility in improving diagnostics and treatment to alleviate disease burden. Compared to conventional mRNA-based transcriptional analysis, the use of miRNA-based biomarkers offers several advantages [124]. Firstly, miRNAs are more stable than mRNA and hence can be isolated from challenging samples such as FFPE blocks more easily and enabling larger retrospective studies [125]. Additionally, the increased stability enables minimally invasive detection of miRNAs in bodily fluids such as blood and urine. Lastly, miRNAs offer high specificity for cell/tissue provenance as well as for disease stage and therapeutic response [124]. Given these advantages and the well-established therapeutic and diagnostic potential of miRNAs in other cancers, the role of miRNAs in informing how we engage with KCs clinically is of great importance.

Several studies have investigated the presence of miRNAs in circulation as a potential prognostic biomarker for cSCC. miR-124 levels are significantly decreased in the serum of cSCC patients compared to healthy controls [75], while miR-10a levels are increased in cSCC patients [56]. Balci et al. [126] compared the miRNA content in the plasma of cSCC and BCC patients, finding miR-30a-5p, miR-576-3p, miR-25-3p, and miR-19a-3p to be downregulated, and miR-186-5p, miR-875-5p, miR-30c-3p, and miR-145-5p to be upregulated compared to healthy controls. One study compared the miRNA profiles of organ transplant recipients’ cSCC lesions and serum, finding only miR-1290 and miR-1246 to be significantly upregulated in both tumor and serum compared to control tissue/serum [55]. This valuably indicates that some miRNA expression patterns can be shared in both tumor and in serum, with potential utility as an accessible biomarker for disease progression and prognosis. With the exception of miR-10a whose expression correlates with poorer prognosis [56], none of these miRNAs have been used to understand diagnosis or prognosis to date. Given circulating miRNAs are easily accessible and stable, without the need for invasive biopsies of tumor tissue, their potential as biomarkers warrants sustained investigation.

Prognostic analyses have, however, been assessed using miRNAs sourced directly from tumors. Rock et al. [39] demonstrated that miR-21-5p and miR-31-5p expression is upregulated only in cSCC lesions on aggressive sites of the body, specifically on or near the ears and lips, as opposed to the trunk and extremities where tumors tend to be less aggressive. miR-21 was again implicated in cSCC prognosis, found to be highly expressed in invasive cSCC but not cSCC in situ [40]. Separately, miR-205 and miR-221 expression correlate with local recurrence and positive surgical margins, respectively [104,127].

It is important to note that these studies generally represent isolated reports of clinical utility for miRNA dysregulation in KCs, and the current lack of consensus (Figure 1 and Figure 2) demands further investigation. Given the potential of biomarkers to affect treatment decisions in the clinic, rigorous and larger scale studies are imperative before implementation of these findings can be considered. Additionally, the field of miRNA therapeutics remains unexplored in KCs and should be explored in the future.

Collectively, these findings may demonstrate some clinical utility in predicting how individual tumors will behave based on their miRNA profile. However, further larger scale investigations will be necessary to evaluate whether any such measures provide better predictive information than existing clinicopathological measures to have any real-world benefit.

## 6. Conclusions and Future Directions

This review sought to comprehensively summarize the body of literature surrounding miRNA dysregulation in keratinocyte carcinomas. We conclude that while several miRNAs are indeed implicated in the progression and pathogenesis of both cSCC and BCC, the clinical utility of this knowledge is, at present, minimal. By highlighting existing strengths and shortcomings of the literature, we anticipate that this review will serve as a guide to refine the focus of future studies in this area. Specifically, implementation of updated methodology as suggested throughout this review can help unify standards and facilitate translation of the results into clinical practice.

Future efforts to integrate these findings with existing genomic and transcriptomic understanding of KCs, especially of metastatic disease, will be of significant benefit to further unravel the molecular complexity of these diseases. Furthermore, we recommend the exploration of miRNAs as prognostic and therapeutic biomarkers in KCs as well as elucidation of the role of exosomal miRNAs as regulators of the TME in KCs. Additionally, the use of miRNAs has not yet been explored in KCs but should be considered as a valid therapeutic approach.

## Figures and Tables

**Figure 1 ijms-25-08493-f001:**
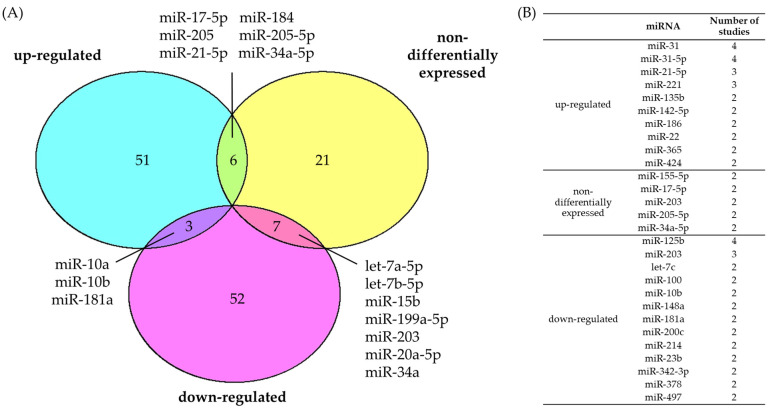
Consensus of dysregulated miRNAs in cSCC. (**A**) Conflicting evidence leaves the deregulation status of several miRNAs unclear. The intersections of Venn diagram represent miRNAs whose differential expression trend has been conflictingly reported in two or more studies. (**B**) The table presents dysregulated/non-differentially expressed miRNAs based on the consensus of two or more studies regarding their dysregulation status. Only comparisons of cSCC to healthy skin were included in these analyses.

**Figure 2 ijms-25-08493-f002:**
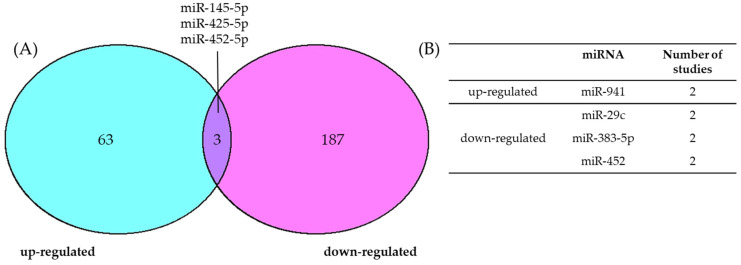
Consensus of upregulated and downregulated miRNAs in BCC. (**A**) Conflicting evidence leaves the deregulation status of several miRNAs unclear. The intersections of the Venn diagram represent miRNAs whose differential expression trend has been conflictingly reported in two or more studies. (**B**) The table presents up- and downregulated miRNAs based on the consensus of two or more studies regarding their dysregulation status. Only comparisons of BCC to healthy skin were included in these analyses.

**Table 3 ijms-25-08493-t003:** Experimental validation of upregulated and downregulated miRNAs in BCC compared to healthy skin.

	miRNA	Tissue Comparison	Cell or Animal Model	Validated Target	Functional Effect	Ref.
**up**	miR-18a	BCC vs. HS	A431	nd	Proliferation ↑ Migration ↑ Cell cycle progression ↑ Apoptosis ↓ Autophagy ↓	[114]
**down**	miR-203	BCC vs. HS	Primary human keratinocytes *K5tTA/TREGLI1* mice	*JUN*	Proliferation ↓ Cell cycle progression ↓ Tumor growth ↓	[115]
	miR-451a	BCC vs. HS	TE 354.T Primary epidermal keratinocytes	*TBX1*	Proliferation ↓ Cell cycle progression ↓	[116]

HS—healthy skin, nd—not determined, ↑—increase in phenotype, ↓—decrease in phenotype.

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
