# Peer review of "Micro RNA Dysregulation in Keratinocyte Carcinomas: Clinical Evidence, Functional Impact, and Future Directions"

_ijms, 2024, doi:10.3390/ijms25158493_

Round 1

Reviewer 1 Report

Comments and Suggestions for Authors

The manuscript entitled "miRNA Dysregulation in Keratinocyte Carcinomas: Clinical Evidence, Experimental Validation, and Future Directions" reports a comprehensive overview of the expression and functions of miRNA in keratinocyte carcinoma, cSCC and BCC. 

In my opinion, the authors reviewed a substantial amount of literature, therefore I believe that this review can be a good reference for future studies. 

However, some aspects needed to be revised:

- pag 4, line 122-124, I would avoid "we believe..." in the sense that a review should exclude interpretation of the results, but, eventually, compare different works. Therefore, I would rephrase the sentence. 

- pag.4 line 136-147 and pag 9, line 182-212, I would see better a separate paragraph reporting the limitations of the different studies. 

- Table 1 and 2: I would suggest adding a table identifying a biological process (maybe by a graphical approach) where a group of miRNA might have a role, to better visualize the importance of a specific group of miRNAs. From the reader's point of view that might have more impact and be useful. 

Author Response

Comments 1: pag 4, line 122-124, I would avoid "we believe..." in the sense that a review should exclude interpretation of the results, but, eventually, compare different works. Therefore, I would rephrase the sentence. 

Response 1:

As per reviewer’s request, we have rephrased the sentence (now pag 4, line 139-140) to avoid introducing our opinion.

“However, it is more likely that these effects are mediated via an alternate target of miR-125b (i.e., STAT3).”

Comments 2: pag.4 line 136-147 and pag 9, line 182-212, I would see better a separate paragraph reporting the limitations of the different studies. 

Response 2:

The group of studies reported on differ between the paragraphs in question and the studies have differing scope, which brings its separate set of limitations that we have addressed in their respective paragraphs. Hence, we respectfully decline the reviewer’s request.

Comments 3: Table 1 and 2: I would suggest adding a table identifying a biological process (maybe by a graphical approach) where a group of miRNA might have a role, to better visualize the importance of a specific group of miRNAs. From the reader's point of view that might have more impact and be useful. 

Response 3:

We thank the reviewer for their suggestion. However, we regard the implementation as not feasible for several reasons. Firstly, there is no consistent validation across studies for functional equivalence of modification of the miRNA level and the protein level. Hence, we cannot say for certain that the observed functional effect is specific to this target of the miRNA and stating a biological effect with confidence would be premature. Additionally, the targets are not part of one molecular process or pathway and have rather widespread biological roles. Furthermore, there is no consensus across studies regarding the evaluation of functional effects leading to gaps in potential effects. For these reasons, we think it best to stick to the reported functional effect and not to oversimplify the evidence base in the literature by picking a select few studies.

Reviewer 2 Report

Comments and Suggestions for Authors

The abstract provides a clear and concise summary of the study, highlighting the role of miRNAs in keratinocyte carcinomas (KCs) and their clinical implications. This section is well-written and gives a good overview of the study's aims and findings.

The introduction effectively sets the stage for the research by explaining the prevalence of KCs and the role of miRNAs in their pathogenesis. However, the authors should elaborate on the primary differences in molecular biology between BCC and cSCC as described in the literature. Additionally, it would be beneficial to justify the focus on miRNA dysregulation in KCs specifically and to discuss any existing therapeutic strategies targeting miRNAs in other types of cancers that could be relevant to KCs.

The methodology is described clearly, including the search strategy for relevant studies and the inclusion criteria. However, more details are needed on the specific databases used for the literature search and any exclusion criteria applied to refine the search results. Furthermore, it is important to describe how the quality and reliability of the included studies were assessed, and to discuss the reasons for excluding the four retracted studies and how their inclusion might have impacted the review’s conclusions.

The results section provides a detailed account of the dysregulated miRNAs in cSCC and BCC, including consensus findings and experimental validations. The findings are well-presented, but it is important to compare the identified miRNAs in cSCC and BCC in terms of their regulatory targets and functional roles. Additionally, the lack of consensus in miRNA dysregulation findings across different studies should be addressed, and the implications of these findings for future clinical practices or therapeutic developments should be discussed.

The discussion synthesizes the results well, emphasizing the potential clinical applications and the need for further research. However, the main challenges in translating miRNA research findings into clinical applications for KCs should be discussed in more detail. Additionally, the influence of the tumour microenvironment on miRNA expression and function in KCs should be considered. The authors should also propose specific future research directions to address the gaps identified in the current literature.

The conclusion effectively summarizes the key findings and reinforces the need for further research to enhance the clinical utility of miRNA dysregulation knowledge in KCs. This section is well-written and provides a strong ending to the manuscript.

I recommend this manuscript for publication pending revision. The authors should address the questions and suggestions provided to strengthen the manuscript and provide a more comprehensive understanding of miRNA dysregulation in keratinocyte carcinomas.

Comments on the Quality of English Language

English needs minor editing 

Author Response

elaborate on the primary differences in molecular biology between BCC and cSCC as described in the literature. Additionally, it would be beneficial to justify the focus on miRNA dysregulation in KCs specifically and to discuss any existing therapeutic strategies targeting miRNAs in other types of cancers that could be relevant to KCs.

Response 1:

We thank the reviewer for their comment. We think the molecular differences between the diseases is outlined to the depth required for this review. We have however added the following line and additional references for the interested reader to excellent reviews expanding on the topic.

Pag 1, lines 43-44 “For a more detailed account of the molecular differences between cSCC and BCC please refer to references [2,10,11].”

Additionally, we have added the following section on therapeutic strategies involving miRNAs as requested.

Pag 2, lines 60 – 70. “Due to their oncogenic and tumor-suppressive functions, miRNAs have been explored as therapeutic targets in cancer . Recent reviews by Menon et al. [18] and Seyhan [19] provide an excellent overview on the strategies for miRNA-targeting therapeutics and its associated challenges. Briefly, strategies targeting oncogenic miRNAs focus on counteracting its regulatory effect on tumor suppressor genes. For this purpose, antagonistic miRNAs, modified nucleic acids, miRNA sponges, masking miRNAs, and small molecule inhibitors targeting miRNA biogenesis are under investigation. Conversely, strategies involving tumor suppressive miRNAs focus on replenishing miRNA levels by delivering a mimetic or gene therapy. However, there are major challenges associated with use of miRNA therapeutics (i.e., delivery, stability, and toxicity concerns).”

Comments 2: The methodology is described clearly, including the search strategy for relevant studies and the inclusion criteria. However, more details are needed on the specific databases used for the literature search and any exclusion criteria applied to refine the search results. Furthermore, it is important to describe how the quality and reliability of the included studies were assessed, and to discuss the reasons for excluding the four retracted studies and how their inclusion might have impacted the review’s conclusions.

Response 2:

We have added clarifying details to the search methodology as requested.

Pag 2, lines 87-88. “The search was conducted on PubMed and Google Scholar with no further constraints applied to the search results.”

We have added a clarifying comment on the credibility concerns of retracted articles as justification for their exclusion. We do not feel the need for additional rationale as to the reason for not including retracted studies as the act of retraction is itself exclusion from the scientific literature.

Pag 2, lines 93 - 95. “Of note, there were four studies that have been retracted and were consequently excluded from analyses due to credibility concerns. The factors used for assessing quality and reliability are outlined throughout the review (where applicable). No studies were excluded based on quality.”

Comments 3: The results section provides a detailed account of the dysregulated miRNAs in cSCC and BCC, including consensus findings and experimental validations. The findings are well-presented, but it is important to compare the identified miRNAs in cSCC and BCC in terms of their regulatory targets and functional roles. Additionally, the lack of consensus in miRNA dysregulation findings across different studies should be addressed, and the implications of these findings for future clinical practices or therapeutic developments should be discussed.

Response 3:

We have added a paragraph comparing the regulatory and functional impact of miRNAs between BCC and cSCC. However, it should be noted that the scope of such a comparison is very limited due to the paucity of literature on miRNA dysregulation in BCC. Additionally, we have added a paragraph to address our concerns around the lack of consensus and its implications for clinical translatability of the finding (section 5).

Pag 13, lines 401 – 412: “Only one miRNA - miR-203 – demonstrates downregulation with experimental vali-dation across both cSCC and BCC. Though the validated targets differ, with MYC and JUN targeted in cSCC and BCC respectively, these are both recurrently altered oncogenes across many cancers. Functionally, forced overexpression of miR-203 demonstrated similar effects in cSCC and BCC, reducing proliferation, cell cycle progression and tumour growth in cell and xenograft models [43,115]. Given miR-203 dysregulation has been documented across a range of tumour types [121-123], it is unlikely that this observation points to a keratinocyte-specific pathogenesis. Indeed, the lack of consensus surrounding this miRNA in cSCC (Figure 1) complicates interpretation of this finding without further investigation. The otherwise lack of consensus between miRNAs in cSCC and BCC points to distinctive disease progressions and ultimately highlights the importance of tumour-specific investigation to elucidate the role of miRNAs in KCs.”

Pag 15, lines 452 - 457: “It is important to note that these studies generally represent isolated reports of clinical utility for miRNA dysregulation in KCs, and the current lack of consensus (Tables 1 and 2) demands further investigation. Given the potential of biomarkers to affect treatment decisions in the clinic, rigorous and larger scale studies are imperative before implementation of these findings can be considered. Additionally, the field of miRNA therapeutics remains unexplored in KCs and should be explored in the future.”

Comments 4: The discussion synthesizes the results well, emphasizing the potential clinical applications and the need for further research. However, the main challenges in translating miRNA research findings into clinical applications for KCs should be discussed in more detail. Additionally, the influence of the tumour microenvironment on miRNA expression and function in KCs should be considered. The authors should also propose specific future research directions to address the gaps identified in the current literature.

Response 4:

We have amended the conclusions section as highlighted in red text below to include more detailed future directions.

6. Conclusions and Future Directions

This review sought to comprehensively summarize the body of literature surrounding miRNA dysregulation in keratinocyte carcinomas. We conclude that while several miRNAs are indeed implicated in the progression and pathogenesis of both cSCC and BCC, the clinical utility of this knowledge is, at present, minimal. By highlighting existing strengths and shortcomings of the literature, we anticipate that this review will serve as a guide to refine the focus of future studies in this area. Specifically, implementation of up-dated methodology as suggested throughout this review can help unify standards and facilitate translation of the results into clinical practice.

Future efforts to integrate these findings with existing genomic and transcriptomic understanding of KCs, especially of metastatic disease, will be of significant benefit to further unravel the molecular complexity of these diseases. Furthermore, we recommend the exploration of miRNAs as prognostic and therapeutic biomarkers in KCs as well as elucidation of the role of exosomal miRNAs as regulators of the TME in KCs. Additionally, the use of miRNAs has not yet been in explored in KCs but should be considered as a valid therapeutic approach.”

Round 2

Reviewer 2 Report

Comments and Suggestions for Authors

Thank you for responding to my comments and undertaking the requested changes